# OpenReview forum: "Dynamic Risk Assessments for Offensive Cybersecurity Agents"
_NeurIPS.cc/2025/Datasets_and_Benchmarks_Track — NeurIPS 2025 Datasets and Benchmarks Track poster_

### Official Review · Reviewer_PotY · 2025-06-20

**Rating:** 4
**Confidence:** 2

**Summary:**

This paper presents a novel framework for dynamic risk assessment of LLM-based offensive cybersecurity agents. The authors argue that static evaluation methods fall short of characterizing real-world adversarial risk, and propose a more representative threat model in which adversaries are capable of modifying agents along five axes of freedom—repeated sampling, iterative prompt refinement, increasing interaction rounds, self-training, and workflow refinement. Using open-source CTF benchmarks (InterCode, NYU CTF, Cybench), the authors demonstrate that even within a fixed compute budget (e.g., 8 H100 GPU hours), significant capability gains (>40%) can be achieved, underscoring the need to account for such attack surfaces in AI risk evaluations.

**Dataset Code Accessibility:**

Yes

**Dataset Code Comments:**

there is sufficient detail to support reproducibility in the provided code.

**Ethical Considerations:**

No, there are no or only very minor ethics concerns

**Limitations Weaknesses:**

1. The paper does not adequately position itself against recent state-of-the-art systems in offensive cybersecurity agents such as D CIPHER (Udeshi et al., 2025), ReaperAI (Fang et al., 2024), or multi-agent collaborative systems. These works demonstrate high performance on similar CTF tasks and should be explicitly cited and contrasted, especially to highlight the unique value of dynamic risk assessment over architectural novelty or tool expansion.
2. Recent works on workflow-level meta agents (e.g., ADAS, Reflexion, Self-Refine) are only superficially referenced. The discussion would benefit from a clearer taxonomy or comparison to position this paper as a benchmark-style assessment tool rather than another optimization method. There is also a missed opportunity to connect to broader AI safety evaluations such as Occult, CyberSecEval 2, or AutoAdvExBench, which are increasingly used for agent risk auditing.
3. The experiments rely on a single base model (Qwen2.5-32B). While understandable given compute constraints, any additional results or qualitative trends on alternative models (e.g., Claude, GPT, StarCoder) would help to generalize the claims that small compute budgets pose a systemic threat.
4. While the framework is well-structured and code is provided, it is not fully abstracted into a plug-and-play benchmark suite. Ideally, a modular API or protocol to assess arbitrary agents under dynamic threat models would enhance reuse potential and align more strongly with the D&B track’s goals.

**Strengths Contributions:**

1.	The proposed dynamic threat model and risk assessment framework are timely and fill a clear methodological gap in evaluating LLM-based cyber agents.
2.	The evaluation is systematic across three widely used CTF benchmarks and explores the influence of each degree of freedom under constrained resources.
3.	The paper is accompanied by a public codebase and provides implementation details that support reproducibility
4.	The work meaningfully connects technical evaluation to regulatory considerations, referencing SB-1047 and broader discussions around liability and derivative models

---

> ### Author Rebuttal · Authors · 2025-07-31
>
> We thank the reviewer's positive, insightful comments and for recognizing our dynamic risk assessment method fills a clear methodological gap in evaluating cybersecurity. We address the feedback and questions below, and will update the paper accordingly.
>
> ## W1 Missing citations of other cybersecurity agents
> We appreciate the reviewer’s insightful comment and would like to clarify a few points. The primary objective of our study is to introduce a novel evaluation methodology rather than to propose a new agent architecture. While we cited D-CIPHER in our related work (line 357), we acknowledge that other relevant studies were not cited and will add these additional references.
>
>  As stated in lines 357–358, to maintain a clear and coherent narrative, we intentionally excluded multi-agent systems from our current scope due to their increased complexity and resource-intensive evaluation requirements. For instance, assessing the risk escalation associated with repeated sampling becomes significantly more challenging in multi-agent scenarios, as the search space expands substantially when sampling across multiple agents. Additionally, training multi-agent systems remains an active and evolving research direction. However, we believe that our dynamic evaluation framework remains applicable, as the fundamental degrees of freedom available to adversaries remain consistent across various agent systems.
>
> ## W2 Recent works on workflow-level meta agents are only superficially cited
>
> We thank the reviewers for their insightful suggestion. As described in lines 269–279, our evaluation of iterative workflow refinement consistently follows the pipeline used to generate refined agent workflows via ADAS, with detailed prompts provided in Appendix B.11. Regarding iterative prompt refinement, additional experimental details, prompts, and qualitative examples are thoroughly discussed in Appendices B.10 and D.2. We would like to clarify that our primary intention here is to illustrate how adversaries can modify agent systems along different dimensions, rather than to propose novel approaches or compare competing methods within the same direction (e.g., alternative strategies for iterative prompt refinement [1, 2]). To mitigate potential confusion, we will expand our related work section accordingly in the revised manuscript. We remain happy to engage further with reviewers on any additional questions.
>
> Our paper is positioned as addressing gaps in typical cybersecurity risk evaluation methodologies, particularly in policy-relevant settings, rather than introducing a novel dataset— making it orthogonal to efforts focused specifically on benchmark development. Instead of assigning a static value to represent an agent's risk level, we advocate for future risk assessments to explicitly incorporate the dynamics of adversarial modifications under specified compute budgets. This dynamic perspective addresses a critical gap that currently exists within recent cybersecurity agent benchmarks [3, 4, 5]. Furthermore, we believe this dynamic risk-assessment methodology can be seamlessly integrated with existing benchmarks to provide more comprehensive evaluations.
>
>
> [1] Arumugam, Dilip, and Thomas L. Griffiths. "Toward efficient exploration by large language model agents." arXiv preprint arXiv:2504.20997 (2025).
>
> [2] Pryzant, Reid, et al. "Automatic prompt optimization with" gradient descent" and beam search." arXiv preprint arXiv:2305.03495 (2023).
>
> [3] Kouremetis, Michael, et al. "Occult: Evaluating large language models for offensive cyber operation capabilities." arXiv preprint arXiv:2502.15797 (2025).
>
> [4] Bhatt, Manish, et al. "Cyberseceval 2: A wide-ranging cybersecurity evaluation suite for large language models." arXiv preprint arXiv:2404.13161 (2024).
>
> [5] Carlini, Nicholas, et al. "AutoAdvExBench: Benchmarking autonomous exploitation of adversarial example defenses." arXiv preprint arXiv:2503.01811 (2025).
>
> ## W3 Limited Model
>
> We acknowledge that we are unable to run more experiments due to the limited compute budget. In order to make our study accurate and well grounded, although the maximum compute budget we evaluate is merely 8 GPU hours, we repeated each experiment at least 5 times and took the average, by doing so the total GPU hours consumed for evaluation are way beyond 8 GPU hours.
>
> Take Figure 6(b) as an example, here we want to show the comparative analysis under the fixed compute budget in non-stateful environments. Following the data provided in Appendix B.7, we detail the compute cost for plotting this figure below:
> 1. Repeated sampling: we repeated our evaluation with N=20 for 35 times, each repetition consumes 1.12 GPU hours → 39.2 GPU hours in total.
> 2. Increase max rounds of interactions:
>     1. We repeated our evaluation with N=30 for 17 times, each repetition consumes 1.85 GPU hours → 31.45 GPU Hours
>      2. We repeated our evaluation with N=40 for 10 times, each repetition consumes 2.53 GPU Hours → 25.3 GPU Hours
>      3. We repeated our evaluation with N=50 for 5 times, each repetition consumes 3.61 GPU Hours → 18.05 GPU Hours
>      4. We repeated our evaluation with N=60 for 5 times, each repetition consumes 4.9 GPU Hours → 24.5 GPU Hours
>      5. We repeated our evaluation with N=70 for 5 times, each repetition consumes 6.87 GPU Hours → 34.35 GPU Hours
>      6. We repeated our evaluation with N=80 for 5 times, each repetition consumes 7.96 GPU Hours → 39.8 GPU Hours
> 3. Iterative prompt refinement: we repeated our evaluation from k=1 to k=15 for 5 times, each repetition consumes 8.02 GPU Hours → 40.1 GPU Hours
> 4. Self-Training: we repeated our evaluation on the checkpoint fine-tuned with 5 epochs for 5 times, each repetition consumes 1.12 GPU Hours, plus the cost for fine-tuning (5.98 GPU Hours) → 11.6 GPU Hours
> 5. Iterative workflow refinement: We repeated our evaluation on the workflow derived from iteration 2 (since it shows improvement compared to the base scaffolding) for 5 times, each repetition consumes 1.38GPU hours, plus the cost for agent workflow optimization (5.76 GPU Hours) → 12.66 GPU Hours
>
> To sum up, the total compute cost is 277.01 GPU Hours. We will add more experiment details to the appendix in our revised version.
>
> Meanwhile, at the time we submitted our paper, open-weight LLMs generally struggled with performing agentic tasks effectively, primarily due to limited proficiency in using tools and managing long-context inputs—a shortcoming echoed by recent research [1, 2]. Qwen-2.5-32B-Coder stands out as a state-of-the-art open-weight coding model, clearly demonstrating escalating risk trends across all five degrees of freedom. This is particularly evident with self-training, which relies heavily on generating initial successful trajectories to form training datasets.
>
> Consistent with the threat model described in lines 112–126, our analysis primarily focuses on open-weight LLMs, as adversaries retain full control over modifications in such environments. Conversely, closed-weight models accessed via APIs introduce uncertainties related to content moderation for prompting or fine-tuning, as well as potential backend reliance on external web sources. This ambiguity makes it challenging to clearly attribute the risk specifically to the model, rather than to external factors.
>
>
> [1] Zhang, Andy K., et al. "Cybench: A framework for evaluating cybersecurity capabilities and risks of language models." arXiv preprint arXiv:2408.08926 (2024).
>
> [2] Abramovich, Talor, et al. "EnIGMA: Interactive Tools Substantially Assist LM Agents in Finding Security Vulnerabilities." Forty-second International Conference on Machine Learning.
>
> ## W4 Not a plug-and-play benchmark suite
>
> We appreciate the reviewers’ valuable suggestions and recognize the potential for enhancing the benchmark suite. Our current codebase offers scripts enabling convenient one-click evaluation of each degree of freedom. However, it remains difficult to create a fully universal plug-and-play benchmark because agent systems and datasets often differ in their input format requirements. Despite this challenge, the code and resources provided with our paper should facilitate easy adaptation to evaluate other agent systems once the data is converted accordingly.

---

> > ### Author Response · Authors · 2025-08-04
> >
> > Dear reviewer, we hope that our response to your comments has addressed any concerns or questions. If any aspects of our rebuttal remain unclear or require further clarification, please feel free to let us know. We would be happy to provide additional information or elaborate as needed. Thanks!

---

> > ### Comment · Reviewer_PotY · 2025-08-05
> >
> > The authors’ response has addressed some of my concerns. However, I still have reservations regarding the generalizability of the proposed approach. While I understand that Qwen is already a state-of-the-art model, I would appreciate it if the authors could also evaluate their method on a less powerful open-source model. It would be ideal if some results or insights could be provided during the rebuttal period. If this is not feasible, please provide a clear justification.

---

> > ### Author Response · Authors · 2025-08-08
> >
> > Dear Reviewer,
> >
> > Thank you for your thoughtful follow-up comment. We add a new set of experiments on Intercode-CTF-Test using Gemma-3-12B-It, and list our results below.
> >
> > ## 1. Repeated sampling and Increasing Max Round of Interactions
> > $N=10$
> > | $k$     | 1     | 2     | 3     | 4     | 5     | 6     | 7     | 8     | 9     | 10    |
> > |--------|-------|-------|-------|-------|-------|-------|-------|-------|-------|-------|
> > | Pass@$k$ | 0.381 | 0.471 | 0.520 | 0.552 | 0.574 | 0.591 | 0.604 | 0.613 | 0.621 | 0.628 |
> >
> > $N=20$
> > | $k$      | 1     | 2     | 3     | 4     | 5     | 6     | 7     | 8     | 9     | 10    |
> > |--------|-------|-------|-------|-------|-------|-------|-------|-------|-------|-------|
> > | Pass@$k$ | 0.419 | 0.514 | 0.561 | 0.587 | 0.603 | 0.615 | 0.624 | 0.631 | 0.636 | 0.640 |
> >
> > $N=30$
> > | $k$      | 1     | 2     | 3     | 4     | 5     | 6     | 7     | 8     | 9     | 10    |
> > |----------|-------|-------|-------|-------|-------|-------|-------|-------|-------|-------|
> > | Pass@$k$ | 0.438 | 0.539 | 0.583 | 0.607 | 0.623 | 0.633 | 0.640 | 0.645 | 0.649 | 0.652 |
> >
> > ## 2. Iterative Prompt Refinement
> > We set $N=20$, use the same prompt as in Appendix B.10, and repeat our experiments 5 times. We list the average pass@k values below.
> > | $k$      | 1     | 2     | 3     | 4     | 5     | 6     | 7     | 8     | 9     | 10    |
> > |----------|-------|-------|-------|-------|-------|-------|-------|-------|-------|-------|
> > | Pass@$k$ | 0.419 | 0.520 | 0.611 | 0.631 | 0.660 | 0.687 | 0.688 | 0.688 | 0.695 | 0.708 |
> >
> > ### 3. Self-Training
> > We collected 25 trajectories from the train set of Intercode CTF, converted them into 126 single-turn conversations, and fed them to the same fine-tuning pipeline. We use the same hyperparameters mentioned in Table 4, and vary the number of epochs from 1 to 3.
> >
> > **1 Epoch**
> > | $k$      | 1     | 2     | 3     | 4     | 5     | 6     | 7     | 8     | 9     | 10    |
> > |----------|-------|-------|-------|-------|-------|-------|-------|-------|-------|-------|
> > | Pass@$k$ | 0.364 | 0.449 | 0.489 | 0.515 | 0.532 | 0.546 | 0.555 | 0.562 | 0.570 | 0.575 |
> >
> > **2 Epochs**
> > | $k$      | 1     | 2     | 3     | 4     | 5     | 6     | 7     | 8     | 9     | 10    |
> > |----------|-------|-------|-------|-------|-------|-------|-------|-------|-------|-------|
> > | Pass@$k$ | 0.311 | 0.417 | 0.475 | 0.510 | 0.534 | 0.550 | 0.562 | 0.571 | 0.577 | 0.583 |
> >
> > **3 Epochs**
> > | $k$      | 1     | 2     | 3     | 4     | 5     | 6     | 7     | 8     | 9     | 10    |
> > |----------|-------|-------|-------|-------|-------|-------|-------|-------|-------|-------|
> > | Pass@$k$ | 0.328 | 0.413 | 0.448 | 0.465 | 0.475 | 0.482 | 0.488 | 0.491 | 0.495 | 0.500 |
> >
> >
> > ## 4. Iterative Workflow Refinement
> > We prompted Gemma-3-12B-It for generating workflow refinement 20 times. However, we found it failed to follow the instructions and provide reasonable workflow refinement.
> >
> > ## Observations
> >
> > We have the following observations:
> > - Increasing $k$ and $N$ consistently improves the agent’s cybersecurity performance, even with a relatively small compute budget.
> > - Iterative prompt refinement is more efficient than repeated sampling for exploration under the same number of rollouts.
> > - Gains from self-training and iterative workflow refinement are harder to obtain for weaker models, because:
> >   - For self-training, they often cannot generate enough successful trajectories initially.
> >   - For iterative workflow refinement, they struggle to follow instructions in long contexts with complex rubrics (see Appendix B.11).
> > - Despite this, risk evaluation along these two directions remains important, as stronger open-source models—likely to be deployed by adversaries—still show clear improvement signals in our paper.  Moreover, as model capabilities keep growing, substantial gains across all five degrees of freedom will become increasingly probable.
> >
> >
> > We will include these additional experiments in the final version and hope they address the reviewer’s concern. We are happy to engage further on any follow-up questions.

---

> > > ### Author Response · Authors · 2025-08-09
> > >
> > > Dear reviewer, we hope that our response to your comments has addressed any concerns or questions. As the author-reviewer discussion period is coming to an end, if any aspects of our rebuttal remain unclear or require further clarification, please feel free to let us know. We would be happy to provide additional information or elaborate as needed. Thanks!

---

### Official Review · Reviewer_3VeF · 2025-06-20

**Rating:** 5
**Confidence:** 4

**Summary:**

This paper argues that current evaluations of LLMs and LLM-based agents on cybersecurity issues underestimate the risk they pose because they treat the agent as static. The authors expand the threat model where the adversary can invest compute time along a number of degrees of freedom. Using three publicly available CTF datasets, they show that with a modest compute budget, they can raise the success rate via these degrees of freedom. The work contributes a new dynamic risk benchmark protocol, a stratified train/test split of InterCode tasks, and an open-source evaluation framework and cost model.

**Dataset Code Accessibility:**

Partly

**Dataset Code Comments:**

I mentioned it in the weaknesses section that I have doubts that data from this paper can be readily used by the community. However, I do believe that this framework and the insights from it on how to conduct these evaluations are important to the community. I do not believe that the data is documented and in digestible format they the community could use right away. Again though, it is not lost on this reviewer that the goal was to perform this risk assessment, it just feels to me like a borderline fit for this specific track.

**Ethical Considerations:**

No, there are no or only very minor ethics concerns

**Final Justification:**

My chief concern with this paper was the suitability for the track, but after discussion with the authors I agree that providing a new methodological approach for evaluating dynamic vs static methods is of great interest to the community. Therefore, I have upwardly revised my review. I do believe that this is well argued and authors responses to the other reviews are sufficient.

**Limitations Weaknesses:**

My chief weakness with this submission is that it may stretch the D&B track expectations. In my evaluation, I do not believe this to be a new dataset, rather an evaluation protocol and a modest amount of data that would be expected of a main track submission. The code with the datasets used is hosted on Github, but the primary contribution is not a new dataset that is on "hosting sites dedicated to ML datasets" as explained in the call for datasets.

If we evaluate as a benchmark, then there is only a single backbone model that is tested, and may not generalize to different model architectures.

Revision: Given conversation with the authors I have re-evaluated my position in the affirmative.

**Strengths Contributions:**

1. The timeliness and potential impact are high. It is the first compute-aware benchmark of LLM in cyber offensive risk, which this highly relevant.
2. Comprehensive evaluation for each DoF and failure-mode taxonomy is well described. The evaluation covers stateful and non-stateful environments as well as iterative prompt refinement which is a plus. I particularly appreciated the evaluation and discussion around self-training having trade-offs in the diversity of solutions that affect the pass@k scores.
3. The paper is well written and clearly shows actionable insights for practitioners and future research.

---

> ### Author Rebuttal · Authors · 2025-07-31
>
> Dear reviewer, thank you very much for your positive, insightful comments and for recognizing that our dynamic risk assessment method is timely and has important policy implications. We want to make a few clarifications to the fitness our work on D&B track in order to address your concern.
>
> According to the submission guideline in NeurIPS track on datasets and benchmarks, it “welcomes all work on data-centric machine learning research (DMLR) that enable or accelerate ML research, covering ML datasets and benchmarks as well as algorithms, tools, methods, and analyses for working with ML data.”, which means proposing a new dataset or benchmark is not required to qualify for this track. In our work, we proposed a data-centric, compute-aware approach to assess the risk of cybersecurity agents dynamically, which aligns with the submission guide. In fact, there is much data-centric ML research that was accepted previously in the NeurIPS D&B track without proposing new datasets or benchmarks. For example:
> 1. Gröger et al., NeurIPS D&B 2024: Intrinsic Self-Supervision for Data Quality Audits.
> 2. Tschalzev et al., NeurIPS D&B 2024: A Data-Centric Perspective on Evaluating Machine Learning Models for Tabular Data.
> 3. Fawkes et al., NeurIPS D&B 2024: The fragility of fairness: Causal sensitivity analysis for fair machine learning
> 4. Raji et al., NeurIPS D&B 2021: AI and the Everything in the Whole Wide World Benchmark.
> 5. Peng et al., NeurIPS D&B 2021: Mitigating dataset harms requires stewardship: Lessons from 1000 papers.
> 6. Koch et al., NeurIPS D&B 2021: Reduced, Reused and Recycled: The Life of a Dataset in Machine Learning Research.
>
> The reason we write this paper is that we realize that there is a huge gap between static risk assessment and dynamic risk assessment, which does not receive enough attention in the community. As the agent’s capability, especially the ability of self-improvement, keeps growing, assessing its dual-use risk in a traditional, static way will likely underestimate the potential. Therefore, we believe that our paper will have a positive impact on advancing the field of risk assessment for cybersecurity agents and other domains with verifiable environments. We hope future agent safety benchmarks could follow the idea of dynamic risk assessment and jointly consider the potential modifications in both the core model and the agent scaffolding, as we have shown in our paper that the cost of expanding the risk “bubble” is extremely simple and cheap.
>
> We remain happy to engage further with reviewers on any additional questions.

---

> > ### Author Response · Authors · 2025-08-04
> >
> > Dear reviewer, we hope that our response to your comments has addressed any concerns or questions. If any aspects of our rebuttal remain unclear or require further clarification, please feel free to let us know. We would be happy to provide additional information or elaborate as needed. Thanks!

---

> > ### Comment · Reviewer_3VeF · 2025-08-05
> >
> > This argument is well reasoned and given that my chief concern was over suitability, I now agree with your assessment. Focusing on a dynamic evaluation vice a static one should be of interest to the community as a whole. I appreciate the citations to other works that I was able to reference when coming to this conclusion.

---

### Official Review · Reviewer_nY2Q · 2025-07-01

**Ethics Flags:** Safety and security
**Rating:** 4
**Confidence:** 3

**Summary:**

The paper introduces a novel approach to dynamically assess the cybersecurity risks posed by offensive AI agents, focusing on how adversaries can exploit various “degrees of freedom” to self-improve agents without external help. Using Capture-the-Flag (CTF) tasks as evaluation benchmarks, the authors show that with limited compute (8 H100 GPU hours), agents can significantly enhance their performance - by over 40% in some cases. The study stresses the importance of evaluating cybersecurity agents under realistic, adversarial conditions, emphasizing both stateful and non-stateful environments.

**Additional Feedback:**

- Clarification of how the work fits the scope of Datasets and Benchmarks.
- Exploration of additional adversarial tactics, including access to auxiliary tools or external memory, and perhaps human feedback or simulated attacker behavior to better emulate realistic adversaries.
-  Expand evaluation to include defense scenarios, i.e. how defensive agents might adapt in parallel.
- The authors raise the need for a more dynamic risk assessment, but how should this be performed? Could it be formalized into a benchmark standard? How can the paper's findings inform real-world red-teaming of AI systems?

**Dataset Code Accessibility:**

Partly

**Dataset Code Comments:**

- The code is made available at: https://github.com/boyiwei/Dynamic-Risk-Assessment
- Datasets used (InterCode, NYU CTF, CyBench) are public and appropriately cited.
- It's not fully clear how the work fits the scope of the Datasets and Benchmarks track.

**Ethical Comments:**

- The paper demonstrates how relatively simple adversaries can significantly boost agent capabilities for offensive tasks, potentially lowering the bar for malicious use.
- While the threat model is detailed, the paper could better contextualize how to mitigate or monitor the risks introduced by these self-improvement pathways.
 - However, the authors do show awareness of the ethical and policy dimensions, linking their findings to relevant regulatory frameworks and emphasizing that evaluations must reflect dynamic threats.

**Ethical Considerations:**

Yes, there are ethics concerns that require attention by the authors

**Final Justification:**

The paper is concerned with a really important topic, of evaluating the true capabilities of LLM agents on offensive cybersecurity tasks - the authors show that adding some natural degrees of freedom can push these capabilities further than previously though, creating the need for more accurate evaluations. In my view, the main weakness is that the authors didn't introduce a comprehensive benchmark based on their observations. However, I acknowledge that this may be challenging, and the discussion with the authors addressed my other concerns. Therefore, I've decided to increase my score.

**Limitations Weaknesses:**

- The scope of the work is not very clear: the authors do not propose a new dataset or benchmark, but mostly enhance the evaluations of existing benchmarks with further degrees of freedom. How does this fit into the Datasets and Benchmarks track? Moreover, the degrees of freedom the authors utilize are generally well-known in the community.
- It would be helpful to include few commented examples of the complete task and agent workflow in the Appendix, to help readers gain a concrete view of the tasks considered, the agentic workflows and tools used, and agent success / failure cases.
- While the five degrees of freedom are well-motivated, other realistic adversarial strategies (e.g., use of external tools, web search, human-in-the-loop tactics) are not explored.
- While the methodology is robust on CTF tasks, it’s unclear how well the findings generalize to other domains of cybersecurity or real-world systems with stricter time or interaction constraints. Moreover, workflow refinement process shows high variance, and it's unclear how stable or generalizable the improvements are across broader agent architectures.

**Strengths Contributions:**

- The paper introduces a comprehensive and realistic threat model that accounts for key “degrees of freedom” (e.g., repeated sampling, iterative prompt refinement, self-training) available to adversaries, filling a crucial gap in prior cybersecurity evaluations.
- It moves beyond static evaluation, proposing a dynamic risk assessment framework which better reflects how agents may evolve post-deployment.
- The authors demonstrate that substantial performance gains can be achieved with small compute budgets across multiple CTF datasets (InterCode, NYU CTF, CyBench), validating the practical relevance of their threat model.
- They obtain insightful findings, such as for example the efficiency of iterative prompt refinement for adversarial improvement, or the cost-effectiveness of their enhancements.

---

> ### Author Rebuttal · Authors · 2025-07-31
>
> We thank the reviewer's insightful comments and for recognizing the comprehensiveness of our evaluation framework and the dynamic risk assessment method. We address the feedback and questions below, and will update the paper accordingly.
>
> ## W1 The scope of the work is not very clear
>
> According to the submission guideline in NeurIPS track on datasets and benchmarks, it “welcomes all work on data-centric machine learning research (DMLR) that enable or accelerate ML research, covering ML datasets and benchmarks as well as algorithms, tools, methods, and analyses for working with ML data.”, which means proposing a new dataset or benchmark is not required to qualify for this track. In our work, we proposed a data-centric, compute-aware approach to assess the risk of cybersecurity agents dynamically, which aligns with the submission guide. In fact, there is much data-centric ML research that was accepted previously in the NeurIPS D&B track without proposing new datasets or benchmarks. For example:
> 1. Gröger et al., NeurIPS D&B 2024: Intrinsic Self-Supervision for Data Quality Audits.
> 2. Tschalzev et al., NeurIPS D&B 2024: A Data-Centric Perspective on Evaluating Machine Learning Models for Tabular Data.
> 3. Fawkes et al., NeurIPS D&B 2024: The fragility of fairness: Causal sensitivity analysis for fair machine learning
> 4. Raji et al., NeurIPS D&B 2021: AI and the Everything in the Whole Wide World Benchmark.
> 5. Peng et al., NeurIPS D&B 2021: Mitigating dataset harms requires stewardship: Lessons from 1000 papers.
> 6. Koch et al., NeurIPS D&B 2021: Reduced, Reused and Recycled: The Life of a Dataset in Machine Learning Research.
>
> The reason we write this paper is that we realize that there is a huge gap between static risk assessment and dynamic risk assessment, which does not receive enough attention in the community. As the agent’s capability, especially the ability of self-improvement, keeps growing, assessing its dual-use risk in a traditional, static way will likely underestimate the potential.
> Therefore, we believe that our paper will have a positive impact on advancing the field of risk assessment for cybersecurity agents and other domains with verifiable environments.
>
> While certain degrees of freedom we explore are already well-known in the cybersecurity community, incorporating them into the risk assessment of cybersecurity agent systems remains valuable, as existing frameworks currently overlook these considerations (see Table 1 in our paper). Moreover, our use of minimal compute resources combined with straightforward self-improvement strategies specifically highlights how quickly and easily the risk "bubble" can expand.
>
> ## W2 More Detailed Examples on Agent Workflow
> We appreciate the reviewers suggestions and will add more details about the agent workflow used in our work and more qualitative examples in our revised version. We use NYU-CTF Agent as our base agent scaffolding, which contains run_command, createfile, disassemble-decompile and check_flag tools. We provide the structure of CTF challenges in Appendix B.4 / Figure 7, and also provide qualitative examples in appendix D and failure mode analysis in Appendix C. We are more than happy to address any further questions in the follow-up discussions.
>
> ## W3 - Other realistic adversarial strategies are not explored
>
> We acknowledge that real-world adversaries may possess a far greater number of degrees of freedom (DoF) than the five outlined in our paper. This is precisely why we describe adversaries as having “at least” five DoF (see line 54). As noted in lines 119–120, we intentionally exclude adversarial strategies involving external assistance, as incorporating external information could obscure the distinction between risks posed by the model itself and those introduced by external sources. We argue that the five DoF highlighted in our study are the most critical, as they do not rely on external assistance and therefore do not require the adversary to possess cybersecurity expertise. The core objective of our project is to emphasize the need for dynamic risk assessment and to illustrate how the risk “bubble” can expand in a simple and low-cost manner. That said, we agree that future work should broaden the threat model to account for risks arising from the interaction between the model and external sources.
>
>
> ## W4 Generalizability to other domains
>
> We acknowledge the limitations regarding generalization beyond CTF-style tasks and some degrees of freedom may exhibit high variance. Our current experiments were conducted in a controlled CTF-like environment due to its clear task boundaries and evaluability. However, many core challenges—such as limited information access, sparse feedback, and complex reasoning—are shared with broader real-world cybersecurity scenarios (e.g., vulnerability triage, automated reconnaissance and techniques in the MITRE ATT&CK matrix). We note that we already take into account some of these concerns via the stateful setting, where models must improve offline (since they only have a limited shot at the external environment. We will add additional discussion outlining how the workflow abstraction and refinement approach could extend to other security domains, along with caveats around time-critical settings in our revised version.

---

> > ### Author Response · Authors · 2025-08-04
> >
> > Dear reviewer, we hope that our response to your comments has addressed any concerns or questions. If any aspects of our rebuttal remain unclear or require further clarification, please feel free to let us know. We would be happy to provide additional information or elaborate as needed. Thanks!

---

> > > ### Comment · Reviewer_nY2Q · 2025-08-05
> > > **Official Comment by Reviewer nY2Q**
> > >
> > > The authors' response addresses several of my earlier concerns. However, the paper still reads more as "a collection of useful observations" than as an actionable benchmarking framework (and some of these observations, such as the benefits of best-of-k, are already known from prior work).
> > >
> > > Given the scope of the study, it would be particularly valuable if the authors could use their findings to propose a new, actionable benchmarking process for cybersecurity agent evaluation. For instance, prior work might conclude that an agent based model M is harmless on dataset D; yet, under the authors’ proposed benchmarking protocol, one might discover that M can in fact be harmful. Instead of relying on specific models, the authors could design a more general benchmarking scaffold (grounded in the degrees of freedom they identify), into which different models and datasets can be “dropped in” to reveal more representative agent capabilities, and mitigated weaknesses from prior studies.
> > >
> > > In my view, this is the main weakness and area for improvement of the work. For the other points (W2–W4), the authors’ responses largely address my concerns.

---

> > ### Author Response · Authors · 2025-08-06
> >
> > Dear Reviewer,
> >
> > Thank you for your thoughtful follow-up comment. Our current codebase provides scripts designed to enable convenient, one-click evaluation of individual degrees of freedom. Additionally, it allows users to integrate new models and diverse benchmarks, making it a practical playground for red-teamers exploring potential risk expansions along various dimensions. Although not fully plug-and-play—given that agent systems and datasets often have different input format requirements— the code and resources provided with our paper should facilitate easy adaptation to evaluate other agent systems once the data is converted accordingly.
> >
> > Our primary goal is to introduce an actionable benchmarking framework. We emphasize systematically assessing risk across each degree of freedom, enabling comparative analysis across various compute budgets. We also advocate the use of cost-accuracy visualization, similar to Figure 6, to support clear, interpretable insights. We will expand on these actionable benchmarking steps and provide clearer instructions within our codebase in our revised version.

---

> > > ### Comment · Reviewer_nY2Q · 2025-08-06
> > > **Official Comment by Reviewer nY2Q**
> > >
> > > Dear Authors,
> > >
> > > Thank you again for clarifying these points, which make sense to me. I plan to review the code repository in more detail over the next few days, and will consider whether any score adjustment is appropriate.

---

### Official Review · Reviewer_1js7 · 2025-07-03

**Rating:** 4
**Confidence:** 3

**Summary:**

Previous benchmarks for evaluating offensive cybersecurity agents mostly focus on static evaluation. This paper aims to bridge the gap and systematically study the risk of offensive cybersecurity agents. Specifically, this paper introduces a dynamic risk assessment framework under an expanded threat model, where an adversary can leverage a fixed compute budget to enhance an agent's performance without external knowledge. Experimental results demonstrate that even a small compute budget (8 H100 GPU hours) can improve an agent's success rate by over 40%, highlighting the need for dynamic and more realistic risk assessments.

**Dataset Code Accessibility:**

Yes

**Ethical Considerations:**

No, there are no or only very minor ethics concerns

**Final Justification:**

The author's rebuttal has addressed most of my concerns. However, since I am not directly working towards this direction. I would like to keep my score as weak accept.

**Limitations Weaknesses:**

1. I am not working in this research direction, but I have some doubts about the suitability of this paper for the dataset and benchmark track, as its contribution to benchmarking appears to be quite limited based on my understanding.
2. The study intentionally uses a small computing budget to demonstrate that significant improvements are accessible. However, the performance curves in Figure 6 are still rising, which could achieve much higher success rates.
3. The evaluation is performed on CTF challenges, which are a valuable but imperfect proxy for real-world cybersecurity operations.

**Strengths Contributions:**

1. This paper makes a clear contribution to the conceptual shift from static to dynamic risk assessment, which is worth exploring.
2. The work is linked to ongoing policy discussions and legal liability standards, which is impactful for policymakers.

---

> ### Author Rebuttal · Authors · 2025-07-31
>
> We thank the reviewer's positive, insightful comments and for recognizing the contribution of the conceptual shift and policy implications of our dynamic risk assessment framework. We address the feedback and questions below, and will update the paper accordingly.
> ## W1- Suitability on Dataset and Benchmark Track
> According to the submission guideline in NeurIPS track on datasets and benchmarks, it “welcomes all work on data-centric machine learning research (DMLR) that enable or accelerate ML research, covering ML datasets and benchmarks as well as algorithms, tools, methods, and analyses for working with ML data.”, which means proposing a new dataset or benchmark is not required to qualify for this track. In our work, we proposed a data-centric, compute-aware approach to assess the risk of cybersecurity agents dynamically, which aligns with the submission guide. In fact, there is much data-centric ML research that was accepted previously in the NeurIPS D&B track without proposing new datasets or benchmarks. For example:
> 1. Gröger et al., NeurIPS D&B 2024: Intrinsic Self-Supervision for Data Quality Audits.
> 2. Tschalzev et al., NeurIPS D&B 2024: A Data-Centric Perspective on Evaluating Machine Learning Models for Tabular Data.
> 3. Fawkes et al., NeurIPS D&B 2024: The fragility of fairness: Causal sensitivity analysis for fair machine learning
> 4. Raji et al., NeurIPS D&B 2021: AI and the Everything in the Whole Wide World Benchmark.
> 5. Peng et al., NeurIPS D&B 2021: Mitigating dataset harms requires stewardship: Lessons from 1000 papers.
> 6. Koch et al., NeurIPS D&B 2021: Reduced, Reused and Recycled: The Life of a Dataset in Machine Learning Research.
>
> The reason we write this paper is that we realize that there is a huge gap between static risk assessment and dynamic risk assessment, which does not receive enough attention in the community. As the agent’s capability, especially the ability of self-improvement, keeps growing, assessing its dual-use risk in a traditional, static way will likely underestimate the potential.
> Therefore, we believe that our paper will have a positive impact on advancing the field of risk assessment for cybersecurity agents and other domains with verifiable environments. We hope future agent safety benchmarks could follow the idea of dynamic risk assessment and jointly consider the potential modifications in both the core model and the agent scaffolding, as we have shown in our paper that the cost of expanding the risk “bubble” is extremely simple and cheap.
>
> ## W2 Small Compute Budget
>
> We understand the reviewer’s point that a limited compute budget might restrict our ability to observe comprehensive growth trends across certain degrees of freedom. While increasing compute resources could provide clearer insights, our central finding remains valid—that risk can escalate considerably even under minimal compute budgets. This highlights why dynamic risk assessment is essential.
>
> We also want to make sure our study is accurate, well-grounded. Therefore, although the maximum compute budget we evaluate is merely 8 GPU hours, we repeated each experiment at least 5 times and took the average, bringing the cumulative cost well beyond this threshold.
>
> Take Figure 6(b) as an example, here we want to show the comparative analysis under the fixed compute budget in non-stateful environments. Following the data provided in Appendix B.7, we detail the compute cost for plotting this figure below:
> 1. Repeated sampling: we repeated our evaluation with N=20 for 35 times, each repetition consumes 1.12 GPU hours → 39.2 GPU hours in total.
> 2. Increase max rounds of interactions:
>     1. We repeated our evaluation with N=30 for 17 times, each repetition consumes 1.85 GPU hours → 31.45 GPU Hours
>      2. We repeated our evaluation with N=40 for 10 times, each repetition consumes 2.53 GPU Hours → 25.3 GPU Hours
>      3. We repeated our evaluation with N=50 for 5 times, each repetition consumes 3.61 GPU Hours → 18.05 GPU Hours
>      4. We repeated our evaluation with N=60 for 5 times, each repetition consumes 4.9 GPU Hours → 24.5 GPU Hours
>      5. We repeated our evaluation with N=70 for 5 times, each repetition consumes 6.87 GPU Hours → 34.35 GPU Hours
>      6. We repeated our evaluation with N=80 for 5 times, each repetition consumes 7.96 GPU Hours → 39.8 GPU Hours
> 3. Iterative prompt refinement: we repeated our evaluation from k=1 to k=15 for 5 times, each repetition consumes 8.02 GPU Hours → 40.1 GPU Hours
> 4. Self-Training: we repeated our evaluation on the checkpoint fine-tuned with 5 epochs for 5 times, each repetition consumes 1.12 GPU Hours, plus the cost for fine-tuning (5.98 GPU Hours) → 11.6 GPU Hours
> 5. Iterative workflow refinement: We repeated our evaluation on the workflow derived from iteration 2 (since it shows improvement compared to the base scaffolding) for 5 times, each repetition consumes 1.38GPU hours, plus the cost for agent workflow optimization (5.76 GPU Hours) → 12.66 GPU Hours
>
> To sum up, the total compute cost is 277.01 GPU Hours. We will add more experiment details to the appendix in our revised version.
>
> ## W3 CTF Challenge is not perfect proxy for real-world cybersecurity operations
> While there’s a gap between CTF challenges and real-world cybersecurity operations, we still believe it could show signals on the trend of growing risk when we increase the compute budgets for adversaries, as it captures the key ingredient we need — a fully verifiable setting –so they still reveal how risk expands when adversaries have more compute. Moreover, while there are some cybersecurity benchmarks that can simulate closer to real-world cybersecurity challenges, many of them are closed to the public [1, 2] or cannot scale to the larger experiment space we require [3, 4]. CTF challenges therefore strike the right balance of accessibility, scalability, and rigor for measuring agent performance within our proposed framework.
>
>
> [1] Rodriguez, Mikel, et al. "A framework for evaluating emerging cyberattack capabilities of ai." arXiv preprint arXiv:2503.11917 (2025).
>
> [2] Kouremetis, Michael, et al. "Occult: Evaluating large language models for offensive cyber operation capabilities." arXiv preprint arXiv:2502.15797 (2025).
>
> [3] Fang, Richard, et al. "Llm agents can autonomously exploit one-day vulnerabilities." arXiv preprint arXiv:2404.08144 (2024).
>
> [4] Zhu, Yuxuan, et al. "CVE-Bench: A Benchmark for AI Agents' Ability to Exploit Real-World Web Application Vulnerabilities." arXiv preprint arXiv:2503.17332 (2025).

---

> > ### Author Response · Authors · 2025-08-04
> >
> > Dear reviewer, we hope that our response to your comments has addressed any concerns or questions. If any aspects of our rebuttal remain unclear or require further clarification, please feel free to let us know. We would be happy to provide additional information or elaborate as needed. Thanks!

---

> > ### Comment · Reviewer_1js7 · 2025-08-05
> > **Official Comment by Reviewer 1js7**
> >
> > The author's rebuttal has addressed most of my concerns. However, since I am not directly working towards this direction. I would like to keep my score as weak accept.

---

### Note · Authors · 2025-08-12

We thank all the reviewers for their engagement and insightful feedback. To close the discussion, we offer the following clarifications and final remarks below:

1. We believe our work provides a framework for risk assessment that takes the dynamics of adversarial modification into consideration, something that is largely not currently done. Specifically, through our analysis of the offensive cybersecurity scenarios, we highlight that static risk assessment may significantly underestimate the risk potential, thus may lead to a false sense of security. The main point of disagreement among reviewers appears to be the fit for the Datasets & Benchmarks track. We believe our work fits exactly the mold of this track since it outlines best practices for evaluation, something that many past papers in the NeurIPS D&B track have done (see references in reviewer responses).
2. Our codebase serves as a flexible playground for exploring risk expansion across multiple degrees of freedom, allowing researchers to swap models and benchmarks with ease. We believe it can support future compute-aware, dynamic risk assessments for agent systems.
3. We want to make sure our study is accurate and well-grounded. Therefore, for a single model studied in our paper, we spent more than 270 GPU hours to plot its compute cost-pass rate curve in Figure 6, and detailed our data processing methods, confidence interval computation in Appendix B. In response to reviewer requests, we also added a set of experiments on Gemma-3-12B-It during the author-reviewer discussion period and will add this result to our future version.
4. For the degrees of freedom studied in our paper, we intentionally pick the ones that can be improved even without external assistance, thus highlighting how simple and cheap that the risk bubble can expand. That said, we do agree that future work should broaden the threat model to account for risks arising from the interaction between the model and external sources.


We hope these remarks address the reviewers’ concerns, and we are grateful for their constructive suggestions.

---

### Decision · Program_Chairs · 2025-09-18

**Decision:**

Accept (poster)

**Comment:**

Main reasons to accept: The paper sets out to evaluate the capabilities of LLM agents on offensive cybersecurity tasks by adopting a dynamic evaluation strategy under an expanded threat model, where an
adversary can leverage a fixed compute budget to enhance an agent's performance without external knowledge. It shows that even a small compute budget (8 H100 GPU hours) can improve an agent's success rate by over 40%, highlighting the need for dynamic and more realistic risk assessments. The main contributions lie in the introduction of a dynamic risk assessment framework, a conceptual shift.  This is viewed by all reviewers as timely and potentially high impact, being the first compute-aware benchmark of LLM in cyber offensive risk.

Main reasons to reject: The evaluation is performed on CTF challenges, which is viewed as an imperfect proxy for real-world cybersecurity operations, thus it's unclear how well findings might generalize to other domains of cybersecurity or real-world systems. There is also some uncertainty about whether this paper should be accepted as a DB track paper, as it is limited to one model and uses one existing dataset, but the argument that the paper presents a new evaluation/assessment methodology also seems valid.

On balance, I am recommending accept, but I understand I may be stretching the bar a bit. Thus I don't mind if this gets bumped down.